**Data Availability Statement:** Data cannot be shared publicly because it is patient data that can possible identify individual patients, which is in

# Long-term outcomes after revascularization in chronic total and non-total occluded coronary arteries: A regionwide cohort study

**Emil Nielsen Holck**[1,2,3]*, **Lars Jakobsen**[1], **Naja Stausholm Winther**[1], **Lone Juul-Hune Mogensen**[1,2], **Evald Høj Christiansen**[1,2]

1 Department of Cardiology, Aarhus University Hospital, Aarhus, Denmark, 2 Institute of Clinical Medicine, Aarhus University, Aarhus, Denmark, 3 Department of Cardiology, Viborg Regional Hospital, Viborg, Denmark

* eh@clin.au.dk

## Abstract

### Background

Understanding the prognostic impact of percutaneous coronary intervention (PCI) in chronic total occlusion (CTO) is crucial for patient management. Previous studies have primarily been studying prognostic impact of successful versus unsuccessful CTO PCI. This study investigated the prognostic impact of successful and unsuccessful percutaneous coronary intervention (PCI) of chronic total occluded coronary arteries (CTO) with non-CTO PCI as reference.

### Methods

Patients treated with PCI from 2009 to 2019 in the Central Region of Denmark were included in a population-based cohort study. We compared successful and unsuccessful CTO PCI with non-CTO PCI. Exclusion criteria was myocardial infarction within 30 days. Primary outcome was difference in a composite major adverse cardio- and cerebrovascular events (MACCE) encompassing all-cause death, any myocardial infarction, stroke, hospitalization for heart failure or revascularization tracked via nationwide registries.

### Results

Of 21,141 screened patients, 10,638 were enrolled: 9,065 underwent non-CTO PCI, 1,300 had successful CTO PCI, and 273 had unsuccessful CTO PCI. Median follow-up time was 5.9 [3.5;9.0] years and 4,750 MACCEs were recorded. Compared to non-CTO PCI, the adjusted MACCE rate for successful CTO PCI was equivalent (Hazard Ratio (HR): 0.98, 95% Confidence Interval (CI): 0.90–1.07, p = 0.71). In contrast, unsuccessful CTO PCI was associated with a higher MACCE rate (HR: 1.22, 95% CI: 1.04–1.43, p<0.01). HR was adjusted for age, body-mass index, previous revascularization, smoking, kidney disease, two or three-vessel disease, left ventricular ejection fraction, diabetes and comorbidities.

compliance with GDPR and danish law data. According to Danish Law we are therefore not allowed to publicly share data. Data are available from the Statistics Denmark Institutional Data Access / Ethics Committee (contact via phone + 45 7841 0188, contact via email forskningsservice@dst.dk) for researchers who meet the criteria for access to confidential data.

**Funding:** ENH: Received scholarship from Aarhus University. Aarhus University did not play any role in the study design, data collection and analysis, decision to publish or prepration of the manuscript. https://health.au.dk/.

**Competing interests:** I have read the journal's policy and the authors of this manuscript have the following competing interests: ENH: Received institutional funds from Asahi Corp., Orbus Neich Corp. and Phillips Corp. EHC: eceived institutional funds from Asahi Corp., Orbus Neich Corp., Phillips Corp. and Abbott Vascular.

**Abbreviations:** ACS, Acute coronary syndrome; CAD, Coronary artery disease; CCS, Chronic coronary syndrome; CTO, Chronic total occlusion; CV, Cardio-vascular; MACE, Major adverse cardiac events; MACCE, Major adverse cardio and cerebrovascular events; MI, Myocardial infarction; OMT, Optimal medical therapy; PCI, Percutaneous coronary intervention; TVR, Target vessel revascularization; WDHR, Western Denmark Heart Registry.

## Conclusions

The pre-specified hypothesis was accepted. Successful CTO PCI was associated with equivalent long-term outcomes as non-CTO PCI, and unsuccessful CTO PCI was identified as a high-risk group associated to worse outcomes.

## Introduction

Coronary artery disease (CAD) remains the leading cause of mortality globally, often presenting as chronic coronary syndrome (CCS) with chronic total occlusions (CTO), defined as a 100% stenosis that has persisted for $\geq$ 3 months [1]. The prevalence of CTO lesions spans from 16% in CAD all-comer populations to 26% in highly selected cohorts [2, 3]. While optimal medical therapy (OMT) is the first-line treatment for CCS, persistent symptoms warrant consideration of revascularization [4]. The prognostic value of CTO revascularization is challenged with randomized trials reporting neutral outcomes, potentially due to insufficient power and selection bias [5–7]. Previous studies have suggested that long-term major adverse cardiac events (MACE) in successfully revascularized CTO patients and non-CTO patients are equal [8, 9]. However, these studies failed to investigate the prognostic effect of unsuccessful CTO PCI, and until now it has only been shown that all-cause mortality is larger than non-CTO patients [10, 11]. Contrarily, registry-based studies without a non-CTO control group suggest that unsuccessful CTO revascularization correlates with increased risk of MACE in comparison to successful CTO revascularization, albeit with concerns of confounding [12]. This study leverages Denmark's high-quality, population-based cohort data to assesses if short- and long-term outcome after PCI differ between CTO and non-CTO patients without acute myocardial infarction (MI).

## Methods

### Research question and hypothesis

What is the short- and long-term outcome after PCI in CTO-patients compared with non-CTO patients, and is the outcome dependent on procedural success? The hypothesis was that successful CTO PCI was not associated to increased risk of MACCE, but unsuccessful CTO PCI was when comparing with non-CTO patients. The study is reported according to the recommendations in the STROBE statement.

### Study design and population

This study was an observational, region-wide cohort study enrolling all consecutive patients undergoing PCI in the Central Region of Denmark from 2009 to 2019. Patients were eligible for enrolment if registered in the Western Denmark Heart Registry (WDHR). Exclusion criteria was PCI for ST-elevation MI, non-ST-elevation MI, out-of-hospital cardiac arrest (OHCA), or cardiogenic shock within the previous 30 days. Exposure was two-fold and defined as 1) successful CTO PCI or 2) unsuccessful CTO PCI. Patients treated with non-CTO PCI was the reference group. A CTO lesion was defined as a 100% stenosis without any antegrade flow (TIM 0) and the absence of thrombus. As according to the definition of the Academic Research Consortium the occlusion had to have a documented or presumed duration $\geq$ 3 months [1]. Successful CTO revascularization was defined as thrombolysis in myocardial infarction III flow and < 50% residual stenosis of all CTO-lesions. Staged (investment)

procedures in the CTO arm were defined as patients undergoing one or more CTO PCIs within ninety days of the index procedure unless acute coronary syndrome (ACS) was the indication. Baseline variables were obtained from the WDHR, the Danish National Prescription Registry and the Danish National Patient Registry where all data are prospectively obtained after each procedure and/or admission in the Danish National Health Care System. Baseline pharmacological treatment was defined as any prescription collected by the patient within the previous six months before index procedure.

## Outcome

Primary endpoint was major adverse cardiac and cerebrovascular events (MACCE) and consisted of all-cause death, any MI, stroke, hospitalisation for heart failure and any repeat revascularization. Secondary endpoints were the individual components of MACCE, cardiovascular (CV) death, non-CV death, and target vessel revascularization (TVR). Death was obtained December 1, 2021, from the Danish Cause of Death Registry, and CV death was defined CV death and unknown death [13]. MI, stroke, and hospitalisation of heart failure were obtained from the Danish National Patient Registry on December 31, 2022. Nationwide analysis of Troponin I or T was used during the study period to diagnose MI. Positive predictive values of these diagnoses are very high compared with medical record audit (MI: 97%, stroke: 97% heart failure: 79%) [14, 15]. PCI procedural data was obtained from the WDHR on December 31, 2019. TVR was defined as any revascularization attempt of the target CTO vessel after successful index CTO PCI. In the unsuccessful group all TVR events were adjudicated by medical record audit to assess if it was a staged procedure. If the staged procedure was successful, the patient was allocated to the successful group. TVR of non-CTO vessels treated at baseline were not adjudicated as TVR but as any repeat revascularization and were therefore also included in MACCE.

## Ethics

Our study was registered and approved at the internal database at the Central Region of Denmark (project number: 1-16-02-312-21). Ethics approval is not mandatory for registry-based projects in Denmark (The Committee Law §14). Informed consent was waived by the institutional review board according to Danish law (The committee law §10). Data for this project was obtained May 4th, 2023, and assessed through the servers of Statistics Denmark. Unique personal identification number was anonymized for the researchers and therefore it was not possible to identify individual patients.

## Statistics

Baseline continuous variables are expressed as mean (± standard deviation) or median [interquartile range] dependent on distribution, and categorical variables are expressed as n (%). P-values for descriptive statistics are not reported due to compliance to STROBE statement [16]. The exposure of successful and unsuccessful CTO PCI was compared to non-CTO PCI using Kaplan-Meier cumulative incidence curves and cox-regression for the primary endpoint and all-cause death. Cumulative incidence curves were censored at the 75th percentile of follow-up time [17]. For secondary endpoints the competing risk of death was accounted for by calculating Nelson-Aalen cumulative incidence curves and the Fine and Gray proportional hazards model. The primary and secondary endpoints were followed from the date of the index procedure or the last staged procedure and censored at emigration (loss-to-follow-up), occurrence of an event or December 1, 2021, which ever came first. Landmark analysis for the primary endpoint of MACCE was performed by stratifying follow-up time in 0 to 30 days, 30 to 365

days and 1 to 9 years and this was supported by time-varying event rates using generalized additive modelling with restricted cubic splines [18]. The hazard ratio was adjusted using a backwards selection model where a set of pre-defined risk factors were tested. If a p-value < 0.10 was observed in univariate analysis, the variables were included in the multivariate model. In the multivariate model variables with p-value > 0.05 were excluded one by one starting with the variable with the highest p-value. To avoid missing data for independent variables chained multiple imputation was applied assuming that data were missing at random. Risk of variables not missing at random were assessed using by dependence on exposure and/or outcome. All analysis was performed at Statistics Denmark using STATA 18.

## Results

### Participants

From January 1, 2009, to 31$^{st}$ December 2019, 21.639 patients underwent PCI in the Central Region of Denmark. In total, 498 patients were excluded due to missing lesion data, exposure variable or being non-Danish citizen. In the CTO group 535 (25.3%) were excluded due to a clinical indication of MI, out of hospital cardiac arrest or cardiogenic shock, and this number was 9,968 (52.4%) in the non-CTO group. Technical success rate of CTO treatment was 82.6% leaving 273 patients in the unsuccessful CTO group and 1,300 in the successful CTO group (Fig 1). The non-CTO group consisted of the remaining 9,065 individuals.

### Clinical characteristics

Patients with successful CTO revascularization were younger than non-CTO and unsuccessful CTO PCI (successful CTO 65.8±10.8, non-CTO 67.3±10.9 and unsuccessful CTO 67.4±11.6).

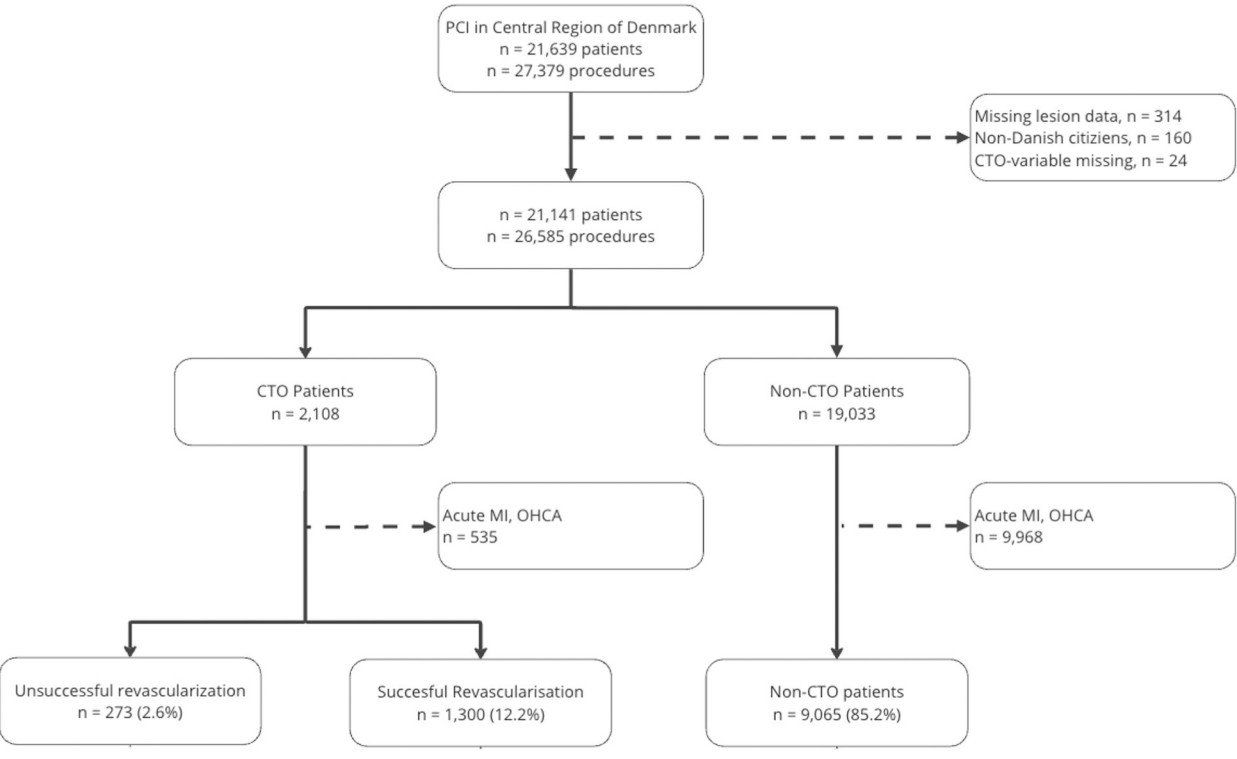

**Fig 1. Flowchart.** Abbreviations: CTO = Chronic Total Occlusion.

In the unsuccessful CTO group, a higher Charlson Comorbidity Index was observed (successful CTO: 3.9±2.3, non-CTO: 3.9±2.3 and unsuccessful CTO: 4.4±2.4) (S1 Table). Cardiac risk factors and medical treatment at baseline are presented in Table 1. Procedural characteristics are reported in Table 2. JCTO score was present in 47.0% in the successful CTO group and 68.4% in the unsuccessful CTO group. Procedural approach was present in 52.8% (S2 Table).

## Outcome

During a median follow-up time of 5.9 [3.5;9.0] years (non-CTO: 6.0 [3.5;9.1], success CTO: 5.7 [3.6;8.9], unsuccessful CTO: 5.6 [2.9;8.6]), 4,006 (44.3%) first time MACCE events occurred in the non-CTO group, 589 (45.3%) in the successful CTO group and 155 (58.3%) in the CTO unsuccessful group. Adjusted MACCE rate was HR (95%CI): 0.98 (0.90;1.07) (p = 0.71) in the successful CTO group, and HR (95%CI): 1.22 (1.04;1.43) (p<0.01) in the unsuccessful CTO group with non-CTO PCI as reference group (Table 3). A higher cumulative incidence of MACCE for unsuccessful CTO compared with non-CTO and successful CTO PCI was observed (Fig 2). The difference was driven by an increased risk of all-cause death (Table 3, S1 Fig). Kaplan-Meier estimates of secondary endpoints are shown in Fig 3. Nineteen CTO patients (successful: 16 (1.2%) and unsuccessful: 3 (1.2%)) had a TVR of a non-CTO vessel treated at baseline, and none of these lesions attributed to the secondary end point of MI.

Within the index admission, 254 (2.8%) in the non-CTO group, 25 (1.9%) in the successful CTO group and 11 (4.1%) in the unsuccessful CTO group experienced a MACCE (p<0.001) (S3 Table). In-hospital death accounted for 64 (0.7%) in the non-CTO group, 1 (0.1%) in the successful CTO group and 1 (0.4%) in the unsuccessful CTO group. TVR in the unsuccessful group occurred in 12 patients (4 non-target-lesion revascularization (TLR), 1 complication, 4 unsuccessful TLR within 12 months, and 3 unsuccessful TLR after 12 months). Variables in the multivariate analysis are reported in S4 Table.

## Landmark analysis

Successful CTO PCI was associated to non-significant lower event rates within the first year (HR (95% CI) 0–30 days: 0.86 (0.67;1.11), 30–365 days: 0.84 (0.69;1.03), 1–9 years: 1.06 (0.95;1.19)), and unsuccessful was only associated to non-significant higher events rates at very short term follow-up and long term (HR (95%CI) 0–30 days: 1.39 (0.94;2.05), 30–365 days: 1.04 (0.73;1.49), 1–9 years: 1.23 (0.99;1.52)) (Fig 4, S2 Fig). Sensitivity analysis excluding revascularizations from the MACCE endpoint found a HR (95%CI) for unsuccessful CTO group: 0–30 days: 1.67 (1.10;2.54), 30–365 days: 1.21 (0.81;1.83) and 1–9 years: 1.30 (1.06;1.61)), indicating that higher MACCE is highly driven by more repeat revascularizations in the non-CTO group (S5 Table).

## Sensitivity analysis of MACCE

Complete case analysis, exclusion of staged patients and adjustment for treatment year only caused minor changes in the point estimate (S5 Table). Adjusting for a propensity score calculated from the full co-variate list resulted in a HR 95%CI in the successful group of 0.93 (0.86;1.02) and 1.21 (1.02;1.42) in the unsuccessful group. Analysing CCS patients only found an adjusted HR (95% CI) 1.00 (0.91;1.10) in the successful CTO group and 1.19 (0.99;1.45) in the unsuccessful group indicating that the indication only had minor impact on the estimate. No Interaction with sex and age was found. Excluding repeat revascularization from MACCE yielded a HR (95%CI) of 1.02 (0.93;1.12) in the successful CTO group and 1.34 (1.13;1.59) in the unsuccessful CTO group. Restricting the analysis to patients with complete

**Table 1. Baseline characteristics.**

| | Non-CTO | CTO Successful | Unsuccessful | Missing |
|---|---|---|---|---|
| | **9065** | **1300** | **273** | |
| Age | 67.3 (10.9) | 65.8 (10.8) | 67.6 (11.5) | 0 (0%) |
| Sex (male) | 6609 (73.0%) | 1031 (79.3%) | 213 (78.6%) | 0 (0%) |
| Body Mass Index | | | | 476 (4.4%) |
| < 18.5 | 111 (1.3%) | 10 (0.8%) | 0 (0.0%) | |
| 18.5–25 | 2528 (29.2%) | 336 (26.9%) | 80 (31.4%) | |
| > 25 | 6015 (69.5%) | 903 (72.3%) | 175 (68.6%) | |
| Heart Failure | 1506 (16.6%) | 292 (22.5%) | 73 (26.7%) | 0 (0%) |
| Left Ventricular Ejection Fraction | | | | |
| < 40 | 987 (12.0%) | 232 (19.0%) | 49 (20.0%) | 974 (8.9%) |
| 40–49 | 536 (6.5%) | 107 (8.8%) | 31 (12.7%) | |
| > 50 | 6675 (81.4%) | 881 (72.2%) | 165 (67.3%) | |
| Previous stroke | 681 (7.5%) | 90 (6.9%) | 18 (6.6%) | 0 (0%) |
| Previous myocardial infarction | 2337 (25.8%) | 371 (28.5%) | 83 (30.4%) | 0 (0%) |
| Previous PCI | 2053 (22.9%) | 488 (37.9%) | 86 (32.1%) | 121 (1.1%) |
| Previous CABG | 452 (5.4%) | 131 (10.7%) | 24 (9.9%) | 743 (6.8%) |
| Smoking | | | | 1231 (11.3%) |
| Active | 2156 (26.8%) | 285 (24.3%) | 51 (21.4%) | |
| Previous | 4018 (50.0%) | 578 (49.3%) | 122 (51.3%) | |
| Never | 1869 (23.2%) | 309 (26.4%) | 65 (27.3%) | |
| Familiar ischemic heart disease | 3830 (47.1%) | 542 (47.5%) | 118 (49.8%) | 1.192 (10.9%) |
| Hypercholesterolaemia | 7253 (80.1%) | 1146 (88.2%) | 220 (81.2%) | 0 (0%) |
| Hypertension | 5634 (63.8%) | 808 (63.7%) | 190 (73.4%) | 273 (2.5%) |
| Diabetes Mellitus | | | | 0 (0%) |
| Non-insulin dependent | 1013 (11.2%) | 167 (12.8%) | 42 (15.5%) | |
| Insulin dependent | 741 (8.2%) | 102 (7.8%) | 21 (7.7%) | |
| Chronic kidney disease | | | | 657 (6.0%) |
| G1 | 1287 (15.1%) | 198 (15.8%) | 31 (13.0%) | |
| G2 | 3638 (42.8%) | 512 (40.7%) | 93 (39.1%) | |
| G3a | 2125 (25.0%) | 355 (28.2%) | 60 (25.2%) | |
| G3b | 1015 (11.9%) | 129 (10.3%) | 34 (14.3%) | |
| G4 | 300 (3.5%) | 44 (3.5%) | 14 (5.9%) | |
| G5 | 138 (1.6%) | 19 (1.5%) | 6 (2.5%) | |
| Charlson comorbidity index | 3.9 (2.3) | 3.9 (2.3) | 4.4 (2.4) | |
| Medical treatment at baseline | | | | |
| Statin | 6427 (71.0%) | 1092 (84.0%) | 208 (76.8%) | 0 (0%) |
| Other lipid lowering | 320 (3.5%) | 80 (6.2%) | 11 (4.1%) | 0 (0%) |
| Aspirin | 5771 (63.8%) | 991 (76.2%) | 189 (69.7%) | 0 (0%) |
| Dual anti-platelet therapy | 798 (8.8%) | 297 (22.8%) | 50 (18.5%) | 0 (0%) |
| Beta receptor antagonist | 4164 (46.0%) | 739 (56.8%) | 148 (54.6%) | 0 (0%) |
| Calcium receptor antagonist | | | | 0 (0%) |
| Group 2 | 2804 (31.0%) | 433 (33.3%) | 109 (40.2%) | |
| Group 1 or 3 | 193 (2.1%) | 28 (2.2%) | 7 (2.6%) | |
| Short-acting nitrates | 2943 (32.5%) | 454 (34.9%) | 79 (29.2%) | 0 (0%) |
| Long-acting nitrates | 2301 (25.4%) | 436 (33.5%) | 89 (32.8%) | 0 (0%) |
| Other anti-anginal therapy | 73 (0.8%) | 12 (0.9%) | 10 (3.7%) | 0 (0%) |

(*Continued*)

**Table 1.** (Continued)

| | Non-CTO | CTO Successful | Unsuccessful | Missing |
|---|---|---|---|---|
| | 9065 | 1300 | 273 | |
| ACE-inhibitor or AIIRB | 4505 (49.8%) | 719 (55.3%) | 171 (63.1%) | 0 (0%) |

Abbreviations: CTO = Chronic total occlusion, PCI = Percutaneous coronary intervention, CABG = Coronary Artery Bypass Grafting, ACE = Angiotensin Converting Enzyme, ARIIB = Angiotensin-II Receptor Blocker

revascularization in all stenosis >50% in the non-CTO group showed an adjusted MACCE HR (95%CI): 1.07 (0.97;1.18) in the successful CTO group and HR (95%CI): 1.32 (1.11;1.57) in the unsuccessful CTO group. Excluding patients with failed non-CTO PCI did not change the point estimate significantly (S3 Fig and S5 Table).

**Table 2. Lesion characteristics.**

| | Non-CTO | CTO Successful | Unsuccessful | Missing |
|---|---|---|---|---|
| | 9065 | 1300 | 273 | |
| Staged treatment | | 160 (12.3%) | 26 (9.5%) | 0 (0) |
| Number of procedures | | 1.2 ± 0.3 | 1.1 ± 0.3 | 0 (0) |
| CTO lesions treated | | 1.1 ± 0.3 | 1.1 ± 0.3 | 0 (0) |
| RCA | | 655 (50.4%) | 122 (44.7%) | |
| LXc | | 281 (21.6%) | 75 (27.5%) | |
| LAD | | 366 (28.2%) | 78 (28.6%) | |
| LM | | 38 (2.9%) | 0 (0.0%) | |
| IM | | 23 (1.8%) | 3 (1.1%) | |
| Non-CTO vessels treated | 1.2 ± 0.4 | 0.5 ± 0.7 | 0.3 ± 0.6 | 0 (0) |
| RCA | 3108 (34.3%) | 155 (11.9%) | 21 (7.7%) | |
| LCx | 2335 (25.8%) | 151 (11.6%) | 28 (10.3%) | |
| LAD | 4688 (51.7%) | 199 (15.3%) | 23 (8.4%) | |
| LM | 646 (7.1%) | 64 (4.9%) | 13 (4.8%) | |
| IM | 147 (1.6%) | 19 (1.5%) | 0 (0.0%) | |
| Diseased vessels* | | | | 244 (2.2%) |
| 1 Vessel disease | 4461 (50.2%) | 470 (37.2%) | 80 (30.5%) | |
| 2 Vessel disease | 2610 (29.4%) | 413 (32.7%) | 81 (30.9%) | |
| 3 Vessel disease | 1807 (20.4%) | 381 (30.1%) | 101 (38.5%) | |
| Indication | | | | |
| CCS | 7121 (78.6%) | 1164 (89.5%) | 203 (74.4%) | |
| UAP | 1056 (11.6%) | 41 (3.2%) | 17 (6.2%) | |
| Non Culprit | 219 (2.4%) | 29 (2.2%) | 6 (2.2%) | |
| Other** | 669 (7.4%) | 66 (5.1%) | 47 (17.2%) | |
| Radiation (Gy/cm$^2$) | 44.9 ± 47.3 | 100.3 ± 89.8 | 82.3 ± 80.7 | 0 (0) |
| Fluoro time (min) | 11.3 ± 11.0 | 31.0 ± 26.4 | 24.5 ± 20.4 | 0 (0) |
| Contrast volume (mL) | 94.6 ± 58.5 | 168.9 ± 93.4 | 124.6 ± 93.1 | 0 (0) |
| Number of stents | 1.6 ± 1.2 | 2.5 ± 1.7 | 0.4 ± 0.9 | 0 (0) |
| Complete revascularization | 7373 (81.3%) | 968 (74.5%) | 0 (0.0%) | 0 (0) |

Abbreviations: CTO = Chronic Total Occlusion, RCA = Right Coronary Artery, LM = Left Main, LCX = Left Circumflex artery, IM = Intermediary branch,

CCS = Chronic Coronary Syndrone, UAP = Unstable Angina Pectoris

*Number of vessels previously treated with PCI or with > 50% diameter stenosis

**Elaboration of other is presented in S6 Table.

**Table 3. Event rates.**

| | Events | Unadjusted HR 95% CI | Adjusted HR 95% CI | p |
|---|---|---|---|---|
| MACCE | | | | |
| Non-CTO | 4.006 (44.3%) | - | - | |
| Successful | 589 (45.3%) | 1.03 (0.94;1.12) | 0.98 (0.90;1.07) | 0.71 |
| Unsuccessful | 155 (58.3%) | 1.42 (1.21;1.67) | 1.22 (1.04;1.43) | <0.01 |
| All-cause mortality | | | | |
| Non-CTO | 2.400 (26.5%) | - | - | |
| Successful | 340 (26.2%) | 0.99 (0.88;1.11) | 1.06 (0.95;1.19) | 0.31 |
| Unsuccessful | 109 (40.9%) | 1.67 (1.38;2.02) | 1.54 (1.27;1.87) | <0.01 |
| CV Death | | | | |
| Non-CTO | 1.041 (11.5%) | - | - | |
| Successful | 153 (11.8%) | 0.97 (0.79;1.19) | 0.92 (0.74;1.14 | 0.43 |
| Unsuccessful | 62 (23.3%) | 2.61 (1.98;3.45) | 2.19 (1.64;2.93) | <0.01 |
| Non-CV Death | | | | |
| Non-CTO | 1.335 (14.8%) | | | |
| Successful | 183 (14.1%) | 0.96 (0.82;1.12) | 1.07 (0.92;1.26) | 0.38 |
| Unsuccessful | 47 (17.7%) | 1.28 (0.96;1.72) | 1.25 (0.93;1.67) | 0.14 |
| Any MI | | | | |
| Non-CTO | 937 (10.4%) | | | |
| Successful | 129 (9.9%) | 0.97 (0.80;1.16) | 0.87 (0.72;1.05) | 0.15 |
| Unsuccessful | 21 (7.9%) | 0.74 (0.48;1.15) | 0.64 (0.41;0.99) | 0.04 |
| Stroke | | | | |
| Non-CTO | 503 (5.6%) | | | |
| Successful | 65 (5.0%) | 0.91 (0.70;1.18) | 0.89 (0.68;1.15) | 0.37 |
| Unsuccessful | 20 (7.5%) | 1.32 (0.84;2.08) | 1.21 (0.76;1.90) | 0.42 |
| Hospitalization for HF | | | | |
| Non-CTO | 462 (5.1%) | | | |
| Successful | 96 (7.4%) | 1.47 (1.18;1.83) | 1.28 (1.02;1.61) | 0.03 |
| Unsuccessful | 21 (7.9%) | 1.53 (0.99;2.38) | 1.14 (0.72;1.81) | 0.59 |
| Any Revascularization | | | | |
| Non-CTO | 1342 (14.8%) | | | |
| Successful | 197 (15.2%) | 1.03 (0.89;1.19) | 0.89 (0.76;1.04) | 0.128 |
| Unsuccessful | 32 (12.0%) | 0.78 (0.55;1.11) | 0.69 (0.48;0.97) | 0.035 |
| Target vessel revascularization | | | | |
| Non-CTO | 773 (8.5%) | | | |
| Successful | 125 (9.6%) | 1.16 (0.96;1.39) | 1.00 (0.83;1.22) | 0.98 |
| Unsuccessful | 12 (4.5%) | 0.51 (0.29;0.91) | 0.45 (0.25;0.80) | <0.01 |

Abbreviations: CI = confidence interval, CTO = Chronic Total Occlusion, MACCE = Major Adverse Cardiac and Cerebrovascular Events, CV = cardiovascular, MI = Myocardial Infarction, HF = Heart Failure

## Discussion

This analysis showed that successfully treated CTO patients had the same long-term risk of MACCE as non-CTO patients, and that the risk was increased in unsuccessfully treated CTO patients. The difference in unsuccessful CTO patients was driven both by higher short- and long-term risk. Worse outcome in unsuccessful CTO patients were attributable to fatal events with higher risk of all-cause death and CV death. Unsuccessfully treated CTO patients had a lower risk of MI, any repeat revascularizations and TVR than non-CTO patients.

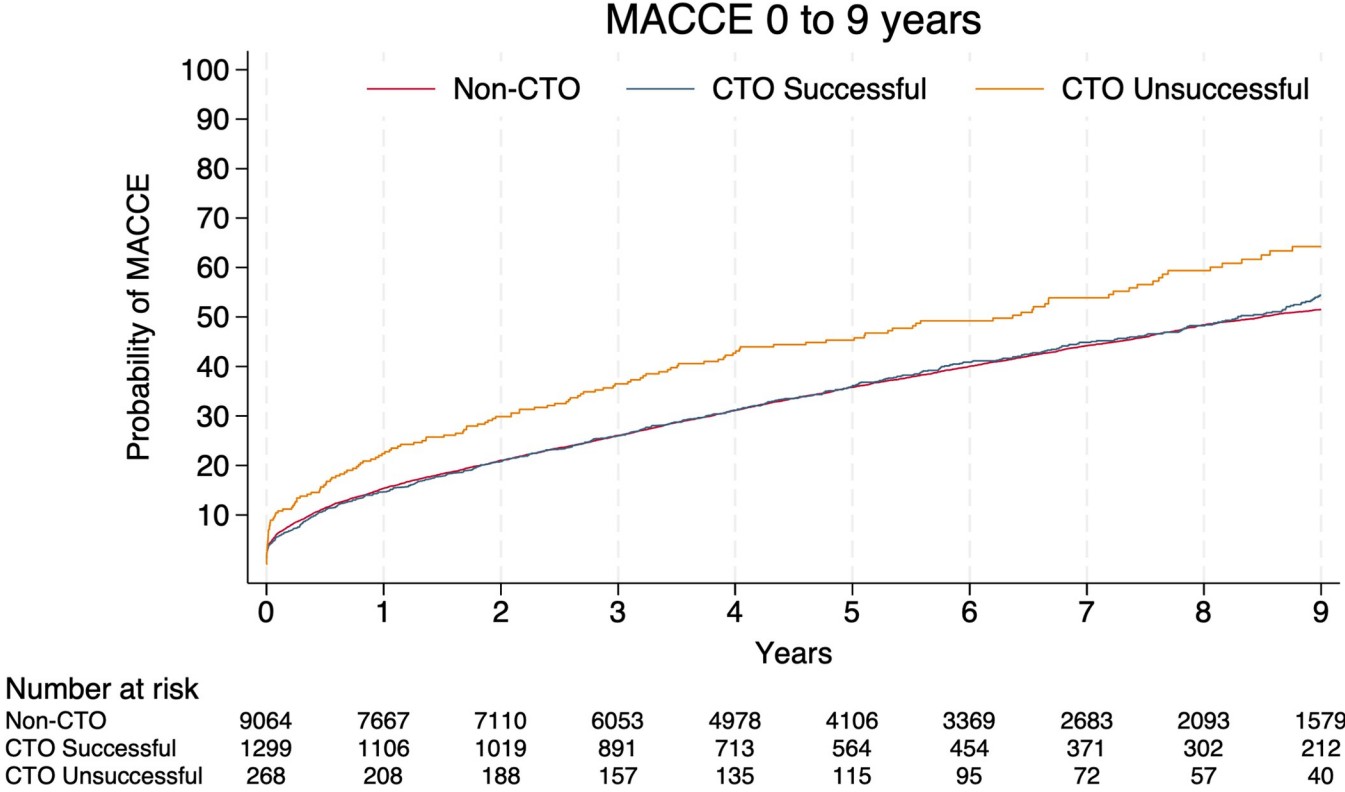

**Fig 2. Cumulative incidence MACCE.** Kaplan-Meier estimate of primary endpoint. Abbreviations: MACCE = Major Adverse Cardiac and Cerebrovascular Events, CTO = Chronic Total Occlusion.

### Prognostic impact of CTO revascularization

Two previous studies have investigated difference in long-term MACE between patients treated for CTO lesions and non-CTO lesions [8, 9]. In line with the findings in our study both Azzalini [9] and Almarzooq [8] observed that successful CTO patients did not have a higher risk of long-term MACE/MACCE than non-CTO patients. Both studies did not investigate unsuccessfully revascularized CTO patients in an adjusted analysis. The findings of our study confirms that successfully revascularized CTO patients have outcome comparable with non-CTO patients. Secondly, we provide a confounder adjusted estimate for patients with unsuccessful CTO treatment in comparison with non-CTO patients as an addon to previous investigation. This supports the findings of previous studies on successful versus unsuccessful CTO PCI which failed to include a non-CTO control group [12]. Thirdly, the long-term results emphasize that the estimate in previous studies may be exaggerated due to short follow-up time. In a meta-analysis conducted by Megaly et al. in 2022 they found that unsuccessful CTO revascularization had a higher risk of MACE than successful CTO revascularization at 12-months (OR (95%CI): 1.64 (1.08;2.44)) [12]. This is supported by the findings in our study where a lower MACCE rate in the successful CTO group than in the unsuccessful CTO group was observed. The effect size in the meta-analysis was much larger than in our study (64% versus 24% increased risk) which may be due to short follow-up time. We observed that within the first 30-days MACCE was higher in the unsuccessful CTO patients, and lower in the successful CTO patients than in non-CTO patients. Our time-dependent landmark analysis emphasizes that the difference in risk between successful and unsuccessful CTO patients is larger within the first year after the procedure than it is from one year and beyond. Therefore,

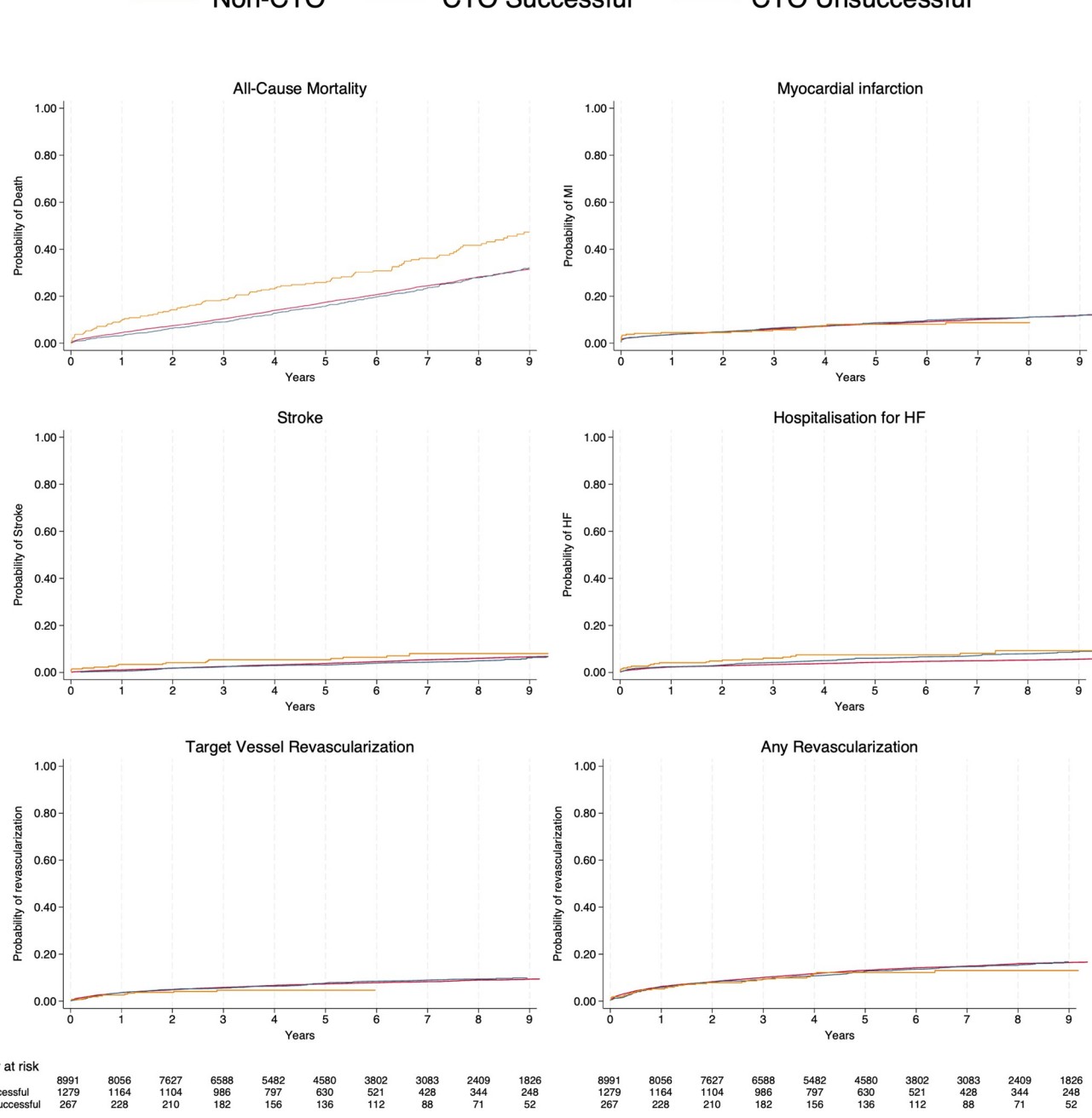

**Fig 3. Individual components of MACCE.** Cumulative incidence plots of secondary endpoints. Abbreviations: CTO = Chronic Total Occlusion, MI = Myocardial Infarction, HF = Heart Failure.

the meta-analysis by megaly et al. that only presents one-year follow-up may show a biased estimate of long-term outcome, and the very long-term outcome presented in this analysis is probably a more valid measure to guide physicians and patients.

An important observation in our analysis was that the individual components of the composite endpoint had diverging impact on MACCE. In a sensitivity analysis we excluded repeat revascularization from MACCE, and worse prognosis in unsuccessful CTO patients was still present. Diverging impact on the composite endpoint may cause an information bias towards

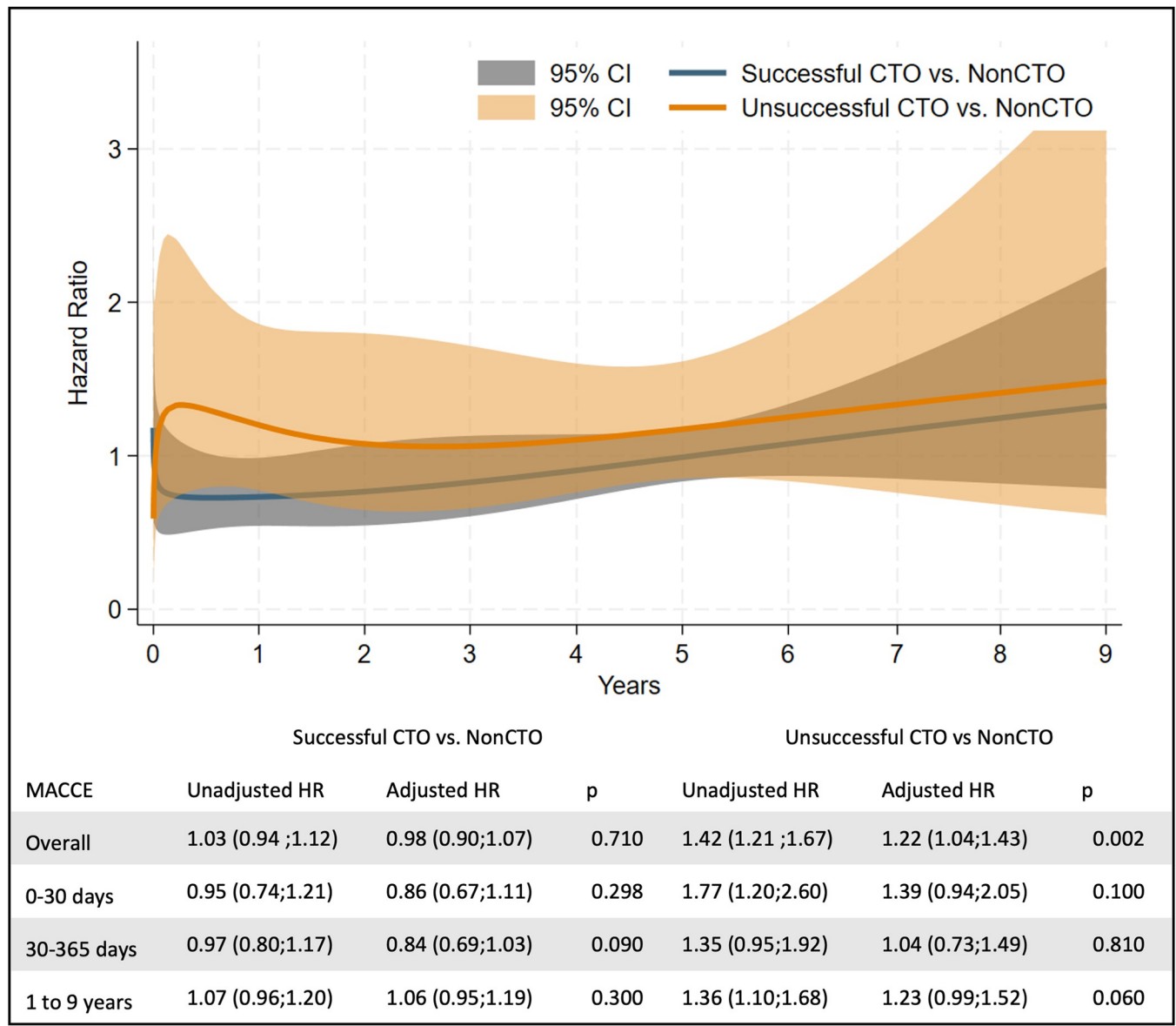

**Fig 4. Landmark analysis.** Upper box: patients enrolled, middle box: time-varying event rates, lower box: landmark analysis using cox regression.
Abbreviations: CTO = Chronic Total Occlusion, MACCE = Major Adverse Cardiac and Cerebrovascular Events, HR = Hazard Ratio.

no difference in previous studies investigating outcome after CTO PCI as the DECISION CTO trial. In DECISION CTO cardiac death was lower (HR (95%CI): 0.56 (0.24;1.34)) in the CTO PCI group, and repeat revascularization was higher (HR (95%CI): 1.34 (0.77;2.31) than in the OMT group [6]. The higher risk of repeat revascularization is likely due to an inherent risk of restenosis in successfully treated CTO patients that is not present in unsuccessfully treated CTO patients. Usage of composite endpoints in randomized controlled trials of CTO revascularization might thus be a possible limitation. Other reasons for failure to prove a prognostic benefit of revascularization in randomized CTO trials may be selection bias, high cross-over rates, and inadequate power [6, 7]. Therefore, it is currently discussed whether CTO patients should be treated as recommended from the landmark CCS trials, with a low fraction of CTO lesions, or be guided by observational data [19, 20]. This highlights the need for observational

data investigating differences in outcome between non-CTO and CTO patients. As we have observed it is reasonable to assume that successfully revascularized CTO patients have a prognosis comparable to non-CTO patients. On the other hand, unsuccessfully treated CTO patients exhibited an increased risk driven primarily by cardiovascular death, which may be attributed to an increased risk of fatal MI or malignant arrhythmias [21, 22] or a detrimental effect of a failed procedure. Patients with fatal MI would be classified as CV-death in our analysis, and this might be the explanation for a lower risk of MI in the unsuccessful group. This is in line with the double jeopardy hypothesis stating that there is an increased risk of OHCA as a complication to MI in patients with a concomitant CTO [21]. The lower risk of spontaneous MI in unsuccessful CTO patients could also be attributed to thrombotic re-occlusions such as stent thrombosis and de-novo lesions outside the stented area in successful CTO patients and non-CTO patients.

## Clinical implications

No matter the reason for the increased risk of death, our analysis identified unsuccessfully treated CTO patients as a high-risk group both in the short- and long-term. The patients included all had a clinical indication for undergoing revascularization attempt. The clinical implication of this main result is that unsuccessfully treated CTO patients require close monitoring and risk factor optimization after a failed attempt. This contrasts with successfully treated CTO patients who can be offered same follow-up as non-CTO patients. Overall, this enables the physician to secure an individualized follow-up treatment of CTO patients. Whether there is a prognostic benefit of CTO revascularization in comparison with OMT remains to be investigated in the ISCHEMIA CTO trial [23]. For now, it is of great importance to improve outcome of patients with a clinical indication for CTO PCI. In line with our findings, it has also been observed that in populations with low procedural success an equally high risk of death is present [24]. Therefore, we suggest implementation of dedicated CTO programs with centralization of complex procedures [25]. Future research should focus on improving technical success and lower procedural complications with e.g. systematic use of staged procedures and pre-procedural planning with computed tomography in combination with dedicated risk scores [26–28].

## Limitations

The study is observational and causal relationships cannot be concluded. Lack of randomization might cause residual confounding and treatment bias favouring the successful group. Likelihood that a CTO was correctly classified as a CTO procedure was 94% in the WDHR, addressing a low risk of including functional CTOs. Missing variables were imputed and did not serve as a major risk of bias in sensitivity analysis. WDHR lacks information on specific CTO parameters as jCTO score and therefore the missing rate is high, yet jCTO was comparable to unsuccessfully revascularized patients in the PROGRESS CTO registry [29]. JCTO is not associated to prognosis and not relevant for confounder adjustment [30]. Secondly, SYNTAX score was not recorded in WDHR. Procedural MI was not routinely assessed, and it was not possible to include this in MACCE. CTO patients treated with OMT are not recorded in the PCI registry and induces a risk of an information bias towards no difference. However, OMT CTO patients would not address the research question investigating procedural outcomes after CTO PCI and would induce a risk of confounding by indication. Including patients with other indications than CCS may induce a selection bias, but sensitivity analysis excluding non-CCS patients did not alter the conclusions. Validity of complications in WDHR is low and not included in the study. However, a full medical record audit of a sub population including 608

patients with complex CTO lesions showed in-hospital event rates comparable to other CTO registries and to the in-hospital MACE rates in this study [31]. Success rate (82.6%) was equal to dedicated CTO-registries (81.4% (75.6;87.1)) and randomized controlled trials (84.5% (76.7;92.4)), and higher than other national registries (63.9% (55.8;72.0)) [32].

## Conclusions

In an unselected cohort of patients treated with PCI for significant CAD successful CTO revascularization was associated to an equal risk of MACCE as patients treated for non-CTO lesions. Unsuccessful CTO revascularization was associated to increased risk of MACCE driven by an excess risk of fatal events, but a lower risk of MI and repeat revascularizations. The findings may improve individualization of treatment and suggests a need for dedicated CTO programs. However, future research should focus on randomizing CTO patients to overcome the inherent risk of observational data.

## Supporting information

**S1 Table. Individual components of Charlson comorbidity index.**
(DOCX)

**S2 Table. JCTO and approach.**
(DOCX)

**S3 Table. In-hospital MACCE.**
(DOCX)

**S4 Table. Variables included in multivariate analysis.**
(DOCX)

**S5 Table. Sensitivity analysis.**
(DOCX)

**S6 Table. Definition of other indication.**
(DOCX)

**S1 Fig. Cardiovascular and non-cardiovascular death.**
(TIF)

**S2 Fig. Cumulative incidence of landmark analysis.**
(TIF)

**S3 Fig. Kaplan-Meier estimates of complete revascularized nonCTO vs. successful and unsuccessful CTO.**
(TIF)

## Acknowledgments

Jakob Hjort at institute for clinical medicine, Aarhus University for assisting with data management.

## Author Contributions

**Conceptualization:** Emil Nielsen Holck, Lars Jakobsen, Evald Høj Christiansen.

**Data curation:** Emil Nielsen Holck, Naja Stausholm Winther, Lone Juul-Hune Mogensen.

**Formal analysis:** Emil Nielsen Holck, Naja Stausholm Winther, Lone Juul-Hune Mogensen.

**Funding acquisition:** Emil Nielsen Holck, Evald Høj Christiansen.

**Methodology:** Emil Nielsen Holck, Lars Jakobsen, Naja Stausholm Winther, Lone Juul-Hune Mogensen, Evald Høj Christiansen.

**Project administration:** Emil Nielsen Holck.

**Resources:** Lars Jakobsen, Evald Høj Christiansen.

**Supervision:** Lars Jakobsen, Evald Høj Christiansen.

**Validation:** Lone Juul-Hune Mogensen.

**Visualization:** Emil Nielsen Holck, Naja Stausholm Winther.

**Writing – original draft:** Emil Nielsen Holck.

**Writing – review & editing:** Emil Nielsen Holck, Lars Jakobsen, Naja Stausholm Winther, Lone Juul-Hune Mogensen, Evald Høj Christiansen.

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
