## [Decision Letter · Decision Letter 0]

21 Feb 2024

PONE-D-24-03357Long-term Outcomes After Revascularization in Chronic Total and Non-Total Occluded Coronary Arteries: A Regionwide Cohort StudyPLOS ONE

Dear Dr. Holck,

Thank you for submitting your manuscript to PLOS ONE. After careful consideration, we feel that it has merit but does not fully meet PLOS ONE’s publication criteria as it currently stands. Therefore, we invite you to submit a revised version of the manuscript that addresses the points raised during the review process. Please submit your revised manuscript by Apr 06 2024 11:59PM. If you will need more time than this to complete your revisions, please reply to this message or contact the journal office at plosone@plos.org. Please include the following items when submitting your revised manuscript:A rebuttal letter that responds to each point raised by the academic editor and reviewer(s). You should upload this letter as a separate file labeled 'Response to Reviewers'.A marked-up copy of your manuscript that highlights changes made to the original version. You should upload this as a separate file labeled 'Revised Manuscript with Track Changes'.An unmarked version of your revised paper without tracked changes. You should upload this as a separate file labeled 'Manuscript'.

We look forward to receiving your revised manuscript.

Kind regards,

Seunghwa Lee

Academic Editor

PLOS ONE

Journal Requirements:

"I have read the journal's policy and the authors of this manuscript have the following competing interests: ENH: Received institutional funds from Asahi Corp., Orbus Neich Corp. and Phillips Corp. EHC: eceived institutional funds from Asahi Corp., Orbus Neich Corp., Phillips Corp. and Abbott Vascular."

Reviewers' comments:

Reviewer's Responses to Questions

**Comments to the Author**

1. Is the manuscript technically sound, and do the data support the conclusions?

Reviewer #1: Yes

Reviewer #2: Yes

Reviewer #3: Yes

Reviewer #4: Yes

Reviewer #5: Yes

2. Has the statistical analysis been performed appropriately and rigorously? 

Reviewer #1: Yes

Reviewer #2: Yes

Reviewer #3: Yes

Reviewer #4: Yes

Reviewer #5: Yes

3. Have the authors made all data underlying the findings in their manuscript fully available?

Reviewer #1: Yes

Reviewer #2: Yes

Reviewer #3: Yes

Reviewer #4: Yes

Reviewer #5: Yes

4. Is the manuscript presented in an intelligible fashion and written in standard English?

Reviewer #1: Yes

Reviewer #2: Yes

Reviewer #3: Yes

Reviewer #4: Yes

Reviewer #5: Yes

5. Review Comments to the Author

Reviewer #1: This manuscript is a large-sample, long-term follow-up observational study. The findings seem to be a little bit interesting, but the statistical methods could be further refined. I think the results would have been more convincing if the authors could have done the propensity score matching analysis, as the number of patients in the three groups currently varies considerably (non-CTO 9065 patients; successful CTO 1300 patients and non-successful CTO 273 patients).

Reviewer #2: 1) It would be good to deliver data conserning SYNTAX Score II, Euroscore, and eventually STS

2) What about investments procedures, did authors include patients after last procedure (if it was successfull or not), sometimes it takes even 3 or 4 times before final decision (procedural failure or success)

3) Table 2 - lesion characteristics - what deas it mean that tere was 1-, 2- and 3- vesse disease? Does it mean that cto was that one which was treated at the end, or there were two CTOs?

4) Neo with meta in unsuccessful group was 2.6 x higher than in the CTO successful group? similar situation was with heart failure? Do the authors think that this could have impacted on MACCE and mortality difference between both groups, which was greater in the unsuccessful group???

Reviewer #3: In this paper, the authors assessed the outcomes of patients undergoing successful and unsuccessful CTO PCI compared to patients with non-CTO PCI. They found that primary outcome of MACCE was higher in unsuccessful CTO PCI group compared to non-CTO PCI group but similar in the successful CTO PCI group compared to non-CTO PCI group. They also performed sensitivity analysis excluding revascularization, and results were similar.

Minior points:

1. Please add how the complete revascularization was defined.

Reviewer #4: In this paper, Holck and colleagues sought to investigate the prognostic impact of unsuccessful CTO PCI in a long term follow-up using a regionwide cohort. We understand that CTO PCI is still a field of debate even after RCT dedicated to this area, once all of RCT published are underpowered for hard endpoints. Thus, good quality registries are important in this gap of knowledge. Some issues regarding this paper are bellow:

1) The novelty (or the lack of novelty) of this paper should be highlighted. Some analyses compering successful versus unsuccessful CTO PCI have been published since 20 years ago and we would like to know how this paper add new information in the gaps about CTO PCI.

2) Nature of this kind of analysis lead to 3 very different populations. I understand that a propensity score matching analysis would be more appropriate than just an adjusted analysis. Could the authors comment?

3) Range of inclusions of patients is wide (2009-2019) and some improvements in PCI technology (2nd generation DES and duration of DAPT, for example) were acquired in this period. Any type os statistical test was performed to investigate the impact of inclusion time (past versus recent) in the results? If not, more details about type of stents would be useful.

4) Definition of MI should be clarified. We know that it has been modified across last 10 years and we do not know if MI in patients enrolled in 2009 is the same as in patients in 2019. It would be useful to understand the lower risk of MI in patients with unsuccessful CTO PCI.

5) I agree with the last sentence in conclusion: These findings suggest that CTO procedures should only be offered in dedicated programs with high success rates. However, given the nature pf this paper, I think that causality is not an answer here. Unsuccessful CTO PCI marks a complex patient, older, with less LVEF, more diabetes and G3 CKD. Thus, I understand that (and this paper do not have power to undo my point) is this myriad of characteristics and not only the unsuccessful procedure the leads to worse prognosis. Finally, high success rates are associated to better prognosis not only because high level techniques, but firstly selecting the more appropriate patient (appropriate indication and technique). The inclusion of CTO patients in OMT (not because unsuccessful PCI) in this analysis could help to understand these remaining questions. But I agree with authors that is not the main question if this paper. I suggest to remove this sentence of the conclusion (at least in abstract) once the eventual reader could misunderstand this speculative point.

Reviewer #5: I have reviewed the manuscript entitled “Long-term Outcomes After Revascularization in Chronic Total and Non-Total Occluded Coronary Arteries: A Regionwide Cohort Study”. In this article, the authors enrolled all consecutive patients undergoing PCI in the Central Region of Denmark from 2009 to 2019. They concluded that successful CTO PCI was associated with equivalent long-term outcomes as non-CTO PCI, and unsuccessful CTO PCI was identified as a high-risk group associated to worse outcomes.

The overall topic of this study is appealing and sought to provide important insights into the long-term outcomes of CTO-PCI. However, some key issues still need to address and consider:

1. The definition of CTO lesion should be more accurate in detail, not only "believed to have persisted for at least three months".

2. The major comparison of this study is non-successful CTO PCI with successful CTO PCI and non-CTO PCI, but how about those success or non-success non-CTO PCI?

3. Why select MACCE but not MACE as the primary endpoint, as the rate of stroke is not expected to be higher in CTO, or non-CTO cohort.

4. What’s the follow-up rate between three arms, which is very crucial for this long-term analysis, and how to deal with those patients with the missing outcomes should be described clearly in the statistics section.

5. The difference was driven by an increased risk of all-cause death and CV death, that hard to explain what the reason of CV death, as the rate of MI and revascularization were lower in non-successful CTO patients.

6. In the result section, the statement looks underdetermined, as the upper and lower bounds of 95%CI were from <1.0 through >1.0. [Successful CTO PCI was associated to lower event rates within the first year (HR (95% CI) 0-30 days:0.86 (0.67;1.11), 30-365 days: 0.84 (0.69;1.03), 1-9 years: 1.06 (0.95;1.19)), and unsuccessful was only associated to higher events rates at very short term follow-up and long term (HR (95%CI) 0-30 days: 1.39 (0.94;2.05), 30-365 days: 1.04 (0.73;1.49), 1-9 years: 1.23 (0.99;1.52))]

7. What’s the definition of MI, including peri-procedural MI and spontaneous MI, what’s the rate of each component during follow-up?

8. The unit of radiation in table 2?

9. Theoretically, all the CTO cases were CCS, how about those UAP patients in table 2, and what’s the definition of “other” indication？

10. We noticed that a number of patients underwent non-CTO PCI, is in the same procedure or staged procedure, what’s the long-term impact of these non-CTO PCIs in CTO patients, What’s the rate of MI and revascularization could be attributed to non-CTO lesion?

6. PLOS authors have the option to publish the peer review history of their article (what does this mean?). If published, this will include your full peer review and any attached files.

Reviewer #1: **Yes: **Peizhi Wang

Reviewer #2: **Yes: **Rafał Januszek

Reviewer #3: **Yes: **Taishi Hirai

Reviewer #4: **Yes: **Eduardo Gomes Lima

Reviewer #5: No

---

## [Decision Letter · Decision Letter 1]

12 Apr 2024

PONE-D-24-03357R1Long-term Outcomes After Revascularization in Chronic Total and Non-Total Occluded Coronary Arteries: A Regionwide Cohort StudyPLOS ONE

Dear Dr. Holck,

Thank you for submitting your manuscript to PLOS ONE. After careful consideration, we feel that it has merit but does not fully meet PLOS ONE’s publication criteria as it currently stands. Therefore, we invite you to submit a revised version of the manuscript that addresses the points raised during the review process.

We look forward to receiving your revised manuscript.

Kind regards,

Seunghwa Lee

Academic Editor

PLOS ONE

Journal Requirements:

Reviewers' comments:

Reviewer's Responses to Questions

**Comments to the Author**

1. If the authors have adequately addressed your comments raised in a previous round of review and you feel that this manuscript is now acceptable for publication, you may indicate that here to bypass the “Comments to the Author” section, enter your conflict of interest statement in the “Confidential to Editor” section, and submit your "Accept" recommendation.

Reviewer #1: All comments have been addressed

Reviewer #2: All comments have been addressed

Reviewer #4: All comments have been addressed

Reviewer #5: All comments have been addressed

2. Is the manuscript technically sound, and do the data support the conclusions?

Reviewer #1: Yes

Reviewer #2: Yes

Reviewer #4: Yes

Reviewer #5: Yes

3. Has the statistical analysis been performed appropriately and rigorously? 

Reviewer #1: Yes

Reviewer #2: Yes

Reviewer #4: Yes

Reviewer #5: Yes

4. Have the authors made all data underlying the findings in their manuscript fully available?

Reviewer #1: Yes

Reviewer #2: Yes

Reviewer #4: Yes

Reviewer #5: Yes

5. Is the manuscript presented in an intelligible fashion and written in standard English?

Reviewer #1: Yes

Reviewer #2: Yes

Reviewer #4: Yes

Reviewer #5: Yes

6. Review Comments to the Author

Reviewer #1: (No Response)

Reviewer #2: (No Response)

Reviewer #4: All the questions were answered adequately.

Statistical issues were clarified.

Speculative sentence in conclusion was removed.

I understand that the manuscript was improved.

Reviewer #5: (No Response)

7. PLOS authors have the option to publish the peer review history of their article (what does this mean?). If published, this will include your full peer review and any attached files.

Reviewer #1: **Yes: **Peizhi Wang

Reviewer #2: **Yes: **Rafał Januszek

Reviewer #4: **Yes: **Eduardo Gomes Lima

Reviewer #5: No

---

## [Author Response · Author response to Decision Letter 1]

26 Apr 2024

It appear that the only questions raised was to review the reference list and upload the figures to PACE. This have been done and did not lead to any changes in the manuscript.

---

## [Editor Report · Decision Letter 2]

4 Jul 2024

Long-term Outcomes After Revascularization in Chronic Total and Non-Total Occluded Coronary Arteries: A Regionwide Cohort Study

PONE-D-24-03357R2

Dear Dr. Holck,

We’re pleased to inform you that your manuscript has been judged scientifically suitable for publication and will be formally accepted for publication once it meets all outstanding technical requirements.

Kind regards,

Giuseppe Andò, M.D., Ph.D.

Academic Editor

PLOS ONE
---

## [Editor Report · Acceptance letter]

5 Jul 2024

PONE-D-24-03357R2 

PLOS ONE

Dear Dr. Holck, 

I'm pleased to inform you that your manuscript has been deemed suitable for publication in PLOS ONE. Congratulations! Your manuscript is now being handed over to our production team.

Kind regards, 

on behalf of

Prof. Giuseppe Andò 

Academic Editor

PLOS ONE